# Breast Cancer Chemotherapeutic Options: A General Overview on the Preclinical Validation of a Multi-Target Ruthenium(III) Complex Lodged in Nucleolipid Nanosystems

**DOI:** 10.3390/cells9061412

**Published:** 2020-06-05

**Authors:** Maria Grazia Ferraro, Marialuisa Piccolo, Gabriella Misso, Francesco Maione, Daniela Montesarchio, Michele Caraglia, Luigi Paduano, Rita Santamaria, Carlo Irace

**Affiliations:** 1Department of Pharmacy, School of Medicine and Surgery, University of Naples “Federico II”, Via D. Montesano 49, 80131 Naples, Italy; mariagrazia.ferraro@unina.it (M.G.F.); marialuisa.piccolo@unina.it (M.P.); francesco.maione@unina.it (F.M.); 2Department of Precision Medicine, University of Campania “Luigi Vanvitelli”, Via L. De Crecchio 7, 80138 Naples, Italy; gabriella.misso@unicampania.it (G.M.); michele.caraglia@unicampania.it (M.C.); 3Department of Chemical Sciences, University of Naples “Federico II”, Via Cintia 421, 80126 Naples, Italy; daniela.montesarchio@unina.it (D.M.); luigi.paduano@unina.it (L.P.)

**Keywords:** breast cancer (BC) therapy, ruthenium complexes, ruthenium-based nanosystems, preclinical studies, cell death pathways, multitarget drugs, triple negative breast cancer (TNBC)

## Abstract

In this review we have showcased the preclinical development of original amphiphilic nanomaterials designed for ruthenium-based anticancer treatments, to be placed within the current metallodrugs approach leading over the past decade to advanced multitarget agents endowed with limited toxicity and resistance. This strategy could allow for new options for breast cancer (BC) interventions, including the triple-negative subtype (TNBC) with poor therapeutic alternatives. BC is currently the second most widespread cancer and the primary cause of cancer death in women. Hence, the availability of novel chemotherapeutic weapons is a basic requirement to fight BC subtypes. Anticancer drugs based on ruthenium are among the most explored and advanced next-generation metallotherapeutics, with NAMI-A and KP1019 as two iconic ruthenium complexes having undergone clinical trials. In addition, many nanomaterial Ru complexes have been recently conceived and developed into anticancer drugs demonstrating attractive properties. In this field, we focused on the evaluation of a Ru(III) complex—named AziRu—incorporated into a suite of both zwitterionic and cationic nucleolipid nanosystems, which proved to be very effective for the in vivo targeting of breast cancer cells (BBC). Mechanisms of action have been widely explored in the context of preclinical evaluations in vitro, highlighting a multitarget action on cell death pathways which are typically deregulated in neoplasms onset and progression. Moreover, being AziRu inspired by the well-known NAMI-A complex, information on non-nanostructured Ru-based anticancer agents have been included in a precise manner.

## 1. Breast Cancer Clinical Classification

The prevention, diagnosis, and treatment of neoplastic diseases are a major problem in developed countries [1]. According to the World Health Organization (WHO), breast cancer (BC) is worldwide one of the most widespread cancer in women; despite the efforts of the scientific community, its occurrence and mortality are expected to increase considerably in the next years [2]. Male BC is not uncommon and must be taken seriously as well, but it accounts for less than 1% of all BC cases [3]. Additionally, a remarkable global upsurge of BC in young women aged between 35 and 40 is underway, with these tumors occurring more aggressive than in older women. Indeed, recent studies provide for BC subtypes as more aggressive and invasive in young women, and current findings suggest that in women aged <45 years, BC is undoubtedly the foremost reason of cancer-associated deaths [4].

BC is an extremely heterogeneous malignancy, completely diverging among distinct patients (intertumor heterogeneity) and even within each individual tumor (intratumor heterogeneity) [5]. The female breast is constituted of milk-producing glands known as lobules, interconnected to each other by ducts. Ducts and lobules are inserted in fatty and connective tissue, rich in blood and lymphatic vessels [6]. Generally, BC originates in cells surrounding the ducts or the lobules. Data suggests there are several BC subtypes that grow and spread (metastatic BC) at different rates, showing different responses to treatments [7]. Moreover, risk factors may broadly vary for each different BC subtype [8]. Besides grouping them according to neoplasm origin (ductal or lobular cancer), to metastasis occurrence (invasive BC) and staging, specific biological markers allow BCs to be clinically divided into distinct subtypes, having different prognostic and therapeutic implications. In particular, genome and transcriptome deep sequencing has led to the identification of at least four major molecular subtypes of invasive BC (luminal A and luminal B hormone receptor positive, HER2-positive, and basal-like) [9]. Among luminal BCs, approximately 80% are ER-positive (estrogen-receptor positive), and about 65% of these are also PR-positive (progesterone-receptor positive) [10]. Luminal A cancers ER/PR-positive and HER-2 negative grow slowly in response to hormone stimulation and so can likely respond to hormone therapy, generally having the best prognosis [11]. Luminal B cancers occur with a quicker growth than luminal A, and their prognosis is slightly worse. They are hormone-receptor positive (estrogen-receptor and/or progesterone-receptor positive), and either HER-2 positive or HER-2 negative, showing high levels of Ki-67 [12]. Tumors overexpressing the human epidermal growth factor receptor-2 (HER-2 positive) tend to be more aggressive than luminal cancers but by chance are responsive to targeted therapy such as treatment with trastuzumab (Herceptin) aimed at the HER-2 protein. About 1 of 5 of BCs are HER-2 positive and hormone-receptor negative, with overexpression of this receptor being behind cellular uncontrolled growth [13]. For the last, basal-type are also known as triple negative breast cancer (TNBC), because these tumors are estrogen receptor negative (ER-), progesterone receptor negative (PR-), as well as HER-2 negative. Although the basal subtype is only found in about 15% of BC diagnoses, it has been shown to be the most aggressive phenotype, unresponsive to treatments and thus with frequently a poor prognosis [14]. Moreover, many BCs associated with BRCA1 gene mutations are TNBC [15]. This type of cancer seems to be at once more widespread among younger and both African and American women [16]. Nevertheless, BC molecular subtypes share several common hallmarks acquired during tumorigenesis process, including uncontrolled proliferation, apoptosis evasion, enhancement of replicative potential, induction of angiogenesis, and tissue invasion, as result of mutations and subsequent metabolic adaptations promoting their survival.

Nowadays, as the field of genomics has developed considerably, researchers can perform fast screening to examine gene sequences alterations with their protein products that might be correlated with precise BC subtypes. Interestingly, many of these gene clusters also mapped onto the four major biological markers described above [17]. Due to continuing advances in the biomedical field, most BCs are identified in time to be profitably cured with surgery, radiotherapy, chemotherapy, or a combination thereof [18]. However, notwithstanding the progress in screening programs and the advancement in the therapeutic field, a considerable amount of BC patients will experience an invasive disease that, to this day, in an important percentage of cases remains incurable thereby justifying the search for new chemotherapeutic strategies.

## 2. Outline on the Therapeutic Approaches for BC Treatment

Currently, different types of treatments are available for patients with BC [19]. In recent years, an upsurge of life-saving treatment advances against BC has been accomplished, thereby creating new optimism. Indeed, there is now an overwhelming choices of treatment options fighting the complex combination of cells in each individual cancer [20]. Here, we introduce in principle the available therapeutic choices for BC treatment, with a brief overview on the most relevant pharmacological ones currently in clinical use. Some treatments are standard (the currently used treatment), and some are being tested in clinical trials [21]. The decisions—surgical procedure, radiotherapy, hormonal (anti-estrogen) therapy, targeted therapy, and/or chemotherapy—can be of critical importance [22]. Treatment for stages I to III BC typically requires surgery and radiation therapy, often with chemo or other drug therapies either before (neoadjuvant) or after (adjuvant) surgery, depending principally on BC spread. Conversely, treatment for stage IV (metastatic BC) is normally a systemic therapy since BC have spread to lymph nodes and to other tissues of the body. Concerning recurrent BC, interventions depend on where the cancer recurs, former treatments and BC type. Recurrence can be local (in the same breast or in the surgery scar), regional (in nearby lymph nodes), or in a distant area [23]. Hormonal therapies are effective just against hormone-receptor-positive BCs; they work by reducing the estrogen levels in the body and/or by inhibiting its stimulatory effect on BC. Of many types are available treatments for hormonal therapy, as well as aromatase inhibitors and selective estrogen receptor modulators [24,25]. As a general rule, cancer targeted therapies are treatments targeting exclusive features of cancer cells, such as proteins (e.g., HER-2) allowing cells to rapidly grow in an abnormal way. Therefore, targeted therapies are generally less harmful than chemotherapy versus healthy cells, including the latest targeted therapies which are of antibody type [26]. For instance, an approved targeted therapy for the ER subtype is the humanized monoclonal antibody trastuzumab (Herceptin), able to block the extracellular domain of HER2, whereas Tamoxifen is the most common clinical drug approved for hormonal therapy in ER-positive breast cancer [27,28]. Immunotherapy can be also employed to handle some types of BCs [29]. Immunotherapeutic drugs targeting specific immune checkpoints can help to restore the immune response against BC, and some of them are facing clinical trial as TNBC treatment options [30].

In addition, a vast repertoire of chemotherapeutic agents is available, e.g., 5-fluorouracil (5-FU), cyclophosphamide, carboplatin, taxanes such as paclitaxel and docetaxel, anthracyclines such as doxorubicin [31,32,33]. Chemotherapeutics correspond to the principal remedies for women whose cancer has become metastatic, but also when it is diagnosed or next to early interventions. Bearing in mind that every BC responds differently to chemotherapy, chemotherapeutics can be used in combination for a specific chemotherapy regimen, both in early-stage BC and in advanced and invasive BC [34]. Although many of these therapeutic options have shown a significant efficacy, the risk of death from BC (30%–35%), as well as the occurrence of cases with natural or acquired chemoresistance, have strongly encouraged the search for alternative therapies targeting [35]. In this context, the exploration of new anticancer treatments that act by triggering multiple cellular responses is currently considered an attractive oncotherapeutic challenge in order to effectively block uncontrolled proliferation [36]. As discussed above, while effective treatments are now available, BC remains an important cause of morbidity and mortality. Besides late diagnosis, along with the disease aggressiveness itself, this is substantially attributable to a merely partial knowledge of the mechanisms behind the failure of conventional therapies and cancer recurrence after quiescence.

## 3. Dysregulated Mechanisms Controlling Apoptosis and Cell Death/Cell Survival in Breast Cancer Cells (BCC): Prospective Targeted Therapies

Cancer can be briefly described as an uncontrolled proliferation of mutant cell clones due to loss of normal controls, resulting in deregulated cell cycle, lack of differentiation, angiogenesis, and local tissue invasion. DNA damage and consequent mutations can also produce devastating effects on the regulation of cell death pathways, thus leading to uninhibited cell proliferation [37,38,39]. In many cases of cancer-related mutations, crucial alterations in the apoptotic machinery causing evasion of programmed cell death are implicated [40]. However, deregulated mechanisms of cell death other than apoptosis may also be involved, such as the autophagic ones [41]. Generally, mutations involve primarily oncogenes and oncosuppressors, finally altering quality and function of protein products that regulate dynamic cellular processes including growth and differentiation, as well as DNA repair. Oncosuppressors code for products which act negatively on cell cycle progression, thus protecting the cell from the accumulation of aberrant mutations. Conversely, oncogenes code for products involved in cell proliferation and survival mechanisms [42]. Similarly, mutations can trigger many instances of cancer chemoresistance able to activate molecular mechanisms that allow to limit drug-induced responses [43]. 

Apoptosis unfolds through two different pathways: the intrinsic pathway—or mitochondria—activated by intracellular signals such as organelles or biomolecular damages (e.g., mitochondrial or DNA injury), and the extrinsic pathway—or death receptor—activated by extracellular signals, both propagated by caspase cascades ultimately leading to cell death [44]. Normally, escape of apoptosis in BCC takes place by definite mechanisms: perturbed signaling of death receptors, deficiency of caspase activity, as well as compromised balance between anti-apoptotic and pro-apoptotic proteins [45]. Bcl-2 protein family regulates the intrinsic pathway and consists of members that either promote (proapoptotic) or inhibit (antiapoptotic) programmed cell death by primarily controlling mitochondrial outer membrane permeabilization (MOMP). By 2008, 25 genes in the Bcl-2 family have been identified [46]. In normal breast epithelial cells, pro-apoptotic and anti-apoptotic signals are closely regulated, while imbalanced apoptotic pathways are often at the “core” of BC onset. Indeed, cell death evasion is currently recognized as a hallmark of cancer. Several findings demonstrate pro-proliferative and anti-apoptotic signals cooperating in initial mammary epithelial cells transformation, where the anti-apoptotic Bcl-2 proteins are considered to play an important role [47]. Consistently, BC spread is faster when correlated to Bcl-2 overexpression, as well as genetic Bax ablation promotes mammary tumor development [48]. In some instances, resistance to targeted BC therapies is also caused by Bcl-2 family proteins. For example, Trastuzumab-resistant HER2-positive BCC commonly upregulate Bcl-2 and downregulate Bax to enhance cell survival [49]. Similarly, following targeted drug regimen interfering with ER signaling in ER-positive cancers, cell survival and chemoresistance seem to be favored by mitochondrial increased activity of Bcl-2 [50]. In addition, Bcl-2 family dysregulation can behave as a key event for resistance to therapies by favoring in turn expression and activity of other pro-survival proteins [51]. Given that tumors growth requires apoptosis escape, and that they commonly share antiapoptotic Bcl-2 family proteins overexpression, targeted inhibition of members of this protein family could represent a valuable therapeutic option. This enlightens the recent advance in BH3-mimetics, a class of small molecule acting as Bcl-2 inhibitors that tightly bind the hydrophobic BH3-binding motif within anti-apoptotic Bcl-2 proteins, promoting the release of apoptotic activators with subsequent MOMP and caspase-dependent cell death [52,53]. However, no current clinical trials are studying the effects of anti-apoptotic Bcl-2 family member inhibition in BC alone. Therefore, several anticancer strategies are ongoing, aiming to restore the physiological balance between cell death and survival by acting on apoptotic pathways. Indeed, targeting the caspase cascade, Bcl-2 family proteins as well as other factors associated with both intrinsic and extrinsic apoptosis signaling, have become among the leading approaches in anticancer therapeutics design [54]. In addition, novel anticancer drugs targeting exclusively specific alterations in cell death signaling pathways could act synergistically with conventional chemotherapeutics in clinical use. In this context, in the last decades many anticancer ruthenium-based candidate drugs have emerged as capable of interacting with proteic targets to exert their anticancer properties [55,56]. Among them, the Ru-based nanomaterials herein discussed and developed as anticancer agents can restore apoptosis in BCC by affecting the expression of some Bcl-2 family members [57]. These outcomes are in line with preclinical studies highlighting the occurrence of apoptosis following ruthenium treatment in cancer cells, through the mitochondria-mediated signal pathway, as well as the death receptor-mediated and/or the endoplasmic reticulum (ER) stress pathways [58]. Therefore, via specific interactions with protein targets such as members of the Bcl-2 family involved in the regulation of apoptosis, ruthenium complexes could selectively reactivate death pathways which are normally suppressed in cancer cells in response to tumorigenic metabolic adaptations promoting cell survival. However, in this framework a breakthrough still needs to be reached, so that no one of the experimentally explored apoptosis-inducing treatments for BC has so far achieved the clinical stages. Thus, in the perspective of new promising anticancer strategies by targeting the apoptosis signaling pathways, many factors should be addressed, as for example to determine whether the observed cytotoxicity in BCC models is comparable in clinical settings.

## 4. Autophagy in BCC: Towards Novel Targets in Anticancer Strategies

Autophagy is fundamentally an evolutionary conserved process ensuring cell survival under stress conditions (e.g., nutrient deprivation, hypoxia, or drugs). Originally termed programmed cell death type II, autophagic cell death differs from apoptosis in being caspase independent. It is accomplished by the coordinated activity of a family of autophagy-related proteins (Atg) and can be initiated via canonical Beclin 1 dependent and non-canonical pathways [59]. Basal levels of autophagy act in healthy conditions as a critical molecular mechanism of homeostasis, clearance of impaired organelles, and nutrient recycling, so as to preserve cellular homeostasis. Alternatively, precise stimuli can promote distinct autophagic mechanisms to address specific stressors [60].

A growing number of data proves that dysregulated autophagy, as well as autophagic pathways failure, play a key role in different malignancies onset and progress, including BC [61,62]. As a double-edged sword, autophagy could either suppress or promote tumorigenesis depending on several dynamics, so its role in cancer is still controversial and the subject of intense research. Different tumor microenvironments further intricate the dual role of autophagy, making its activation a very complicated process to be managed in response to treatments [63]. Accordingly, the impact of autophagy on antitumor therapy has not been demonstrated yet. However, excessive and sustained autophagy leads to cell death since the massive vacuolization of cytoplasm can deprive cell of resources required to survive, acting as a tumor suppressor factor. Several studies demonstrate both canonical and noncanonical autophagic cell death induced in BCC in response to different treatments [64]. Hence, in the last years autophagy potential as an anticancer target has been largely explored. Therapeutic interventions based on the capacity to prompt this biological response are considered as attractive approaches, with the mTOR inhibitors being the best example [65,66]. Nevertheless, targeting the autophagic machinery by chemotherapeutics deserve additional studies, representing an exciting potential molecular mechanism to control cancer cells proliferation. Indeed, numerous anticancer drugs under preclinical studies, both stimulating and inhibiting autophagy, target different steps of this process [67].

Independently form cancer molecular subtypes, BCC that evade therapy-dependent growth inhibition can hypothetically give rise to drug resistant clones, wherein autophagy can be one of the main driver factors in this process. In the case of ER-positive BCs, about the 40% of patients show resistance to targeted anti-estrogen treatments, and this can be frequently correlated with protective autophagy [68]. Remarkably, a significant correlation between ERα and Beclin 1 has been established by recent studies, where the autophagic protein can deactivate the estrogen receptor, making thereby MCF-7 cells less responsive to growth stimulation by estradiol. However, in the same cells, overexpression of Beclin 1 caused resistance to anti-estrogen therapy [69]. In addition, MCF-7 overcoming Tamoxifen effect showed an autophagic flux occurrence with protective function, i.e., interfering with autophagy by pharmacologic intervention can improve Tamoxifen efficacy [70]. Similarly, concurrent treatment of ER-positive T47D BCC with Tamoxifen and autophagic inhibitors caused a significant decrease in cell viability, suggesting autophagy as an important player with pro-survival roles in response to direct ER targeting [71]. Overall, in estrogen deprived models, these findings prove autophagy as involved in mainly cytoprotective function, thus endorsing the use of autophagy inhibitors to make BC more susceptible to anti-estrogenic therapy [72]. Concerning HER-2 enriched BC, acquired resistance to treatments represents one of the main challenges for survival from disease. As in the case of ER-positive BCs, connections among autophagy and HER-2 receptor down-stream signaling have been detected and targeted therapies deactivating HER-2 signaling seem to induce autophagy [73].Chemotherapeutic drugs are largely employed in BC treatment, especially in TNBC where the lack of proper receptors does not allow for targeted therapy. Acquired resistance to treatments can be a major concern and autophagy can be involved as a mechanism allowing cancer to overwhelm sensitivity to therapy [74]. Nevertheless, among compounds with autophagy-regulating activities, we cannot predict autophagy role in a distinct tumor environment [75]. Indeed, as an example, Gemcitabine caused protective autophagy in MDA-MB-231, and cytotoxic autophagy in ER-positive MCF-7 cells [69,76]. In line, preclinical investigations by human TNBC models (MDA-MB-231 and MDA-MB-453) showed cytotoxic autophagy induced by cisplatin (*c*DDP) and cytoprotective autophagy in response to docetaxel [74]. In addition, new drugs alone or in combination can stimulate autophagy-dependent cell death functions. For example, Bcl-2 inhibition by siRNA in MCF-7 cells affects cell growth following Doxorubicin exposure and this occurs via autophagic cell death rather than through enhancement of apoptosis [77]. As well, multitarget Ru(III)-based nanosystems lately developed by our group as anticancer agents induce robust autophagy in some BCC models (either ER-positive or TNBC), which is associated with apoptosis, finally leading to a remarkable antiproliferative effect in vitro [78]. In addition, it was reported that other Ru(II) and Ru(III) derivatives can induce autophagic responses in cancer cells, as proved by the detection of autophagosomes, acidic vesicular organelles (AVOs), and expression of LC3-II [79]. Thus, important correlations have been newly reported concerning ruthenium-based treatments and activation of cytotoxic autophagy. By evaluation of the expression of some key autophagy markers, such as Beclin-1 and LC3-II, a significant upregulation in advanced preclinical models has been clearly detected [80]. In contrast, platinum-based derivatives did not show any pronounced effect, confirming the findings of other studies where cisplatin failed to induce autophagy [81]. According to ruthenium complexes properties, these connections suggest the possibility for Ru-based candidate drugs of novel targets engaged in the regulation of autophagic processes.

Overall, outcomes reveal that in the same disease models different therapeutic modalities can cause activation of distinct types of autophagy, and that autophagy itself can be both cytoprotective and cytotoxic depending by a number of intrinsic and extrinsic factors. Even if autophagy is predominantly recognized as a defensive mechanism allowing cancer cells to escape apoptosis, combination treatments in cells circumventing apoptosis may flow into autophagic cell death, highlighting crosstalk between apoptosis and autophagy as an advanced field to improve cancer treatment [78,81].

## 5. An Outlook on Ru-Based Complexes in the Landscape of Anticancer Metallodrugs in Clinical Trials

During the last decades, several advances in biological targeted approaches for cancer therapy have occurred; however, despite recent progress, the need remains for novel and effective anticancer agents to be available for a variety of clinical cases. In particular, for invasive disease such as TNBC as well as for other metastatic BCs, wide-spectrum and effective chemotherapy agents are still required. Currently, this clinical area is covered only by platinum(II) coordination complexes, including the worldwide approved Cisplatin (*c*DDP), Oxaliplatin and Carboplatin, together with the restricted Nedaplatin and Lobaplatin [33,82]. Of course, absence of any biological selectivity leads to severe off-targets effects especially on high proliferating tissues. Moreover, cancer chemoresistance to platinum treatments is an increasing clinical concern in all chemotherapy regimens [83]. All this has fueled the search for novel advanced therapeutics based on alternative metals and their coordination complexes, showing possibly cellular selectivity but retaining bioactivity versus malignancies as a function of unique mechanisms of action [84]. Given the chemical similarity and within the same metal group, focus on ruthenium complexes has risen noticeably, such that antitumor drugs based on ruthenium are nowadays among the most advanced non-platinum metallodrugs. Ruthenium complexes are in fact found to be striking alternatives to platinum derivatives since they exhibit several advantageous properties matching with rational anticancer drug design and biomedical requests [56,58]. What makes them very attractive drug candidates compared to conventional drugs such as Cisplatin is the ability to interact simultaneously with different cellular biomolecules, including proteins and enzymes over nuclear targets, allowing them to trigger different cell death pathways, as DNA-independent apoptosis. Consequently, the occurrence of multiple targets could also have a critical impact in avoiding the development of chemoresistance which plagues many conventional therapies. Additionally, the combination between ruthenium complexes and available clinical antitumor drugs to synergistically handle tumors is also under intense investigation [85]. With respect to platinum compounds, ruthenium is coordinated at two additional axial positions giving rise mostly to octahedral compounds. Generally, ligands themself and their arrangement modulate the bioactivity of ruthenium complexes, especially concerning their reactivity, hydrophobicity, binding, cellular uptake, and intracellular distribution [86]. Ruthenium compounds exist in three main oxidation states. While Ru(IV) compounds are unstable, Ru(III) complexes have a good thermodynamic and kinetic stability, and can be used as prodrugs, showing antitumor effect by reduction in situ to corresponding Ru(II) counterparts under biological conditions of decreased pH, hypoxia, and increased levels of glutathione, as described later [87,88]. Conversely, Ru(II) can directly inhibit cancer cells via multiple mechanisms, some of which are yet to be elucidated. Indeed, several Ru(II) compounds demonstrated superior antitumor properties than their corresponding Ru(III) counterpart [89]. In addition, combining with their applicability as nanomaterials, in the last years interest in these compounds has further increased [90,91]. The following possibilities are feasible to enhance the water solubility of ruthenium compounds, which in turn is critical for their possible applications in biomedical field: (i) changing the ligand structures; (ii) designing supramolecular ruthenium derivatives; (iii) enclosing ruthenium complexes into nanostructured materials.

Among Ru coordination complexes, NAMI-A, KP1019 and its sodium salt equivalent NKP-1339 are the iconic ones exhibiting structural similarity and having undergone clinical trials (Figure 1) [92,93]. Behaving as prodrugs—being capable of generating active cytotoxic species (Ru II) in situ within specific tumor microenvironments—Ru(III) complexes are endowed with significant antiproliferative activity in addition to a lower toxicity profile. Indeed, an “activation by reduction” mechanism has been proposed for many Ru(III) complexes by a number cellular reductants. Such activation might improve tumor targeting as the reducing and hypoxic cellular microenvironment can favor activation of the metal center, thus improving antiproliferative selectivity for cancer cells [88]. The molecular mechanisms of action at the level of intracellular targets seem to be dissimilar from the DNA-binding mechanism, typically associated with platinum-based drugs. In this context, recent investigations in preclinical models in vitro and in vivo have revealed a range of potential cellular targets other than DNA [94,95]. Different classes of ruthenium compounds retain significant biological activities via interactions with multiple cancer-related protein targets, causing interference in several signaling cascades [96,97]. Indeed, in the last years new targets have emerged for Ru-based anticancer drugs, some preliminary structure-activity relationships can be determined, and docking interactions studies make available reasonable structures for potential protein-metallodrug adducts [55,98].

Unprecedent research on Ru complexes was originally performed by Clarke in the 1980s on simple Ru(III) chloroammine compounds, directly modeled on Cisplatin structure [99]. In 1986 Keppler described firstly the anticancer activity of an original water-soluble anionic Ru(III) complex named KP418, which manifested high efficacy against colorectal cancer [100]. KP418 is the direct precursor of the less toxic indazole analogue KP1019, in turn supplanted by the more soluble sodium salt KP1319/NKP1339//IT-139 [101]. The positive outcomes reported by Keppler and coworkers on Ru(III)-azole complexes prompted in the late 1980s the design of an alternative class of structurally related Ru(III)-DMSO compounds. In this frame, Mestroni and Alessio formulated Ru(III) complexes provided with sulfoxide ligands, showing significant antitumor properties explored on several cancer experimental models [102]. Noteworthy, they also reported about the control of metastatic dissemination from solid tumors by the tested ruthenates [103,104]. The octahedral arrangement around the Ru(III) center of NAMI-A was first conceived by the team of Sava by replacing NAMI as the corresponding chemical stable imidazolium salt. Consistently, NAMI-A revealed a synergistic action in both preventing cell invasion and hampering neo-angiogenesis, thus appearing suitable for metastasis rather than for primary tumors treatment. Indeed, in mice bearing Lewis lung carcinoma, NAMI-A administered *i.p.* considerably decreased lung metastasis weight by about 80%–90% [105,106]. Compared to cisplatin and in line with what said before, a broad variety of biological targets has been revealed for NAMI-A, mainly extracellular rather than nuclear and DNA-based [107]. Therefore, the anti-metastatic capacities of NAMI-A are dependent by its ability to interfere with functions involved in metastasis development, including cell adhesion and migration [108]. Having entered clinical trials in 1999 and reported in 2004, NAMI-A was the first Ru-based drug entering a phase I study performed at the National Cancer Institute of Amsterdam (NKI) on patients suffering different solid tumors [109]. Unfortunately, some side effects were observed and phase II trials using NAMI-A alone were not pursued. In its place, phase II trials were done in combination with gemcitabine in non-small cell lung cancer patients after first line treatment. NAMI-A showed again side effects and was less effective than gemcitabine alone. Due to these negative outcomes, clinical trials were terminated [110].

NKP1339 is currently the most promising Ru(III)-based drug in clinical trials [111]. The original form, KP1019, was revised to improve its aqueous solubility, producing the sodium salt equivalent, NKP1339 [112]. Structurally similar to NAMI-A, NKP1339 is a pro-drug which can bind non-covalently with plasma proteins, especially with albumin through hydrophobic interactions [113]. Indeed, blood proteins adducts formation is more extensive for NKP1339 than NAMI-A; as well, NKP1339 cellular uptake is considered significantly more efficient than the limited one for NAMI-A. Since the complex persists in the pro-drug form before undergoing activation by reduction in target cells following release from albumin, the metal-protein adduct seems not to be involved in the low side effect profile verified throughout the phase I trial [92,93]. DNA is expected to be a primary target for NKP1339, owing for its propensity to accumulate within the nucleus after activation [114]. NKP1339 induces cell cycle arrest in cancer cells, typically within 20÷30 h via activities ascribed to its redox ability. It is in fact able to enhance ROS intracellular production by unsettling redox homeostasis, with consequent upregulation of the pro-apoptotic p38 MAPK pathway, typically stimulated by cellular stress factors, including DNA damage, ROS generation, and cytokines expression, and associated with cell cycle progression [115]. More importantly, this pathway is also implicated in the control of the G1/S and G2/M check points within the cell cycle. Hence, by ROS generation coupled to impaired cellular redox balance, NKP1339 can induce G2/M cell cycle arrest [114]. Concerning cell death pathways activation, most apoptosis develops via the extrinsic pathway. Indeed, whilst mitochondria are among biological targets of NKP1339, the apoptotic induction seems to be orchestrated by either death receptors on cell surface or other mechanisms involving endoplasmic reticulum (ER) homeostasis [116]. Remarkably, cancer overexpression of proteins related with multi-drug resistance (e.g., MRP1, BCRP, LRP, and the transferrin receptor) does not interfere with the drug’s efficacy due likely to its multi-targeting action [117]. During phase I clinical trials, NKP1339 was studied for the treatment of advanced solid tumors. Moreover, studies on patient tolerability, as well as on pharmacodynamic and pharmacokinetic concerns, were performed (Niiki Pharma Inc. and Intezyne Technologies Inc., 2017). The trial (NCT0145297) was successfully completed in 2016 and, as opposed to NAMI-A, demonstrated limited side effects in trial participants [118,119].

To conclude the discussion concerning Ru-based anticancer drugs in clinical studies, in the last years a Ru(II) complex called TLD1433, demonstrating prospective as a photosensitizer for photo-dynamic therapy both in vitro and in vivo, has entered trials [120]. Meanwhile, in the last decades many other Ru complexes endowed with superior anticancer activity have been designed and developed. For some of them, the possibility of entering clinical trials may be not far away [85,98]. Notwithstanding the encouraging outcomes in advanced preclinical and clinical trials, some limitations have been observed for anticancer Ru-based complexes, largely associated to their limited stability in biological environments, in which they can be converted in non-soluble poly-oxo species, impairing both general pharmacokinetic and pharmacodynamic profiles [93,121]. In fact, the discordant note is that only three Ru complexes have reached clinical trials, due to multifactorial concerns (physical, chemical, and biological) [122]. This should be considered in order to ad hoc develop prospective Ru-based drug candidates endowed with the desirable properties in the hope for a future effective anticancer “ruthenotherapy”.

## 6. Ru-Based Drugs Upgrading for Cancer Treatments: Advancements and Prospective Nanostructured Materials

In addition to the compounds ready for clinical appliance, to date numerous Ru-based derivatives, both inorganic and organic, have been investigated throughout advanced preclinical phase [85,87,123]. Indeed, several ruthenium complexes with attractive physico-chemical and biochemical properties, including superior anticancer activity, have been intensively studied as promising candidates within the frame of a prospective “ruthenotherapy” [124,125]. In recent years, many preclinical studies have been conducted on new Ru(II) and Ru(III) scaffolds in different phases of advancement. e.g., the ruthenium complexes RAPTA (ruthenium arene PTA) are a promising class of experimental cancer drugs developed by the Dyson group, exhibiting efficient action in different solid primary tumors. Several derivatives of RAPTA were synthesized, and two of the most notable are RAPTA-C and RAPTA-T, surprisingly showing the ability to inhibit lung metastasis in mice bearing middle cerebral artery mammary carcinoma [126].

With a view to improving stability and anticancer properties of the Ru(III) complex NAMI-A, a class of pyridine-based analogues were later synthesized. Almost simultaneously in 2012, Walsby and coworkers and some of our group reexamined a NAMI-A pyridine derivative, named NAMI-Pyr and AziRu, respectively (Figure 1) [127,128]. In parallel with NAMI-A, this analog holds a pyridine ligand replacing the imidazole group, and sodium replacing imidazolium as the counterion. By virtue of its greater lipophilia, but still maintaining a good solubility in biological microenvironment, AziRu shows per se higher cytotoxicity in vitro than NAMI-A itself. Despite this, in bioscreen in vitro on selected panels of human cancer cells including BBC, bioactivity profile remains rather low, with relatively high IC_50_ values [129]. Probably this is still caused by a slow crossing grade through the cell membranes and a consequent drug low concentration inside the cell, together with possible degradation/instability of the complex in aqueous biophases before reaching molecular targets [130]. However, due to its enhanced physico-chemical properties with respect to NAMI-A, AziRu showed a weak to moderate cytotoxicity against some BCC: as an example, its IC_50_ value is half than that of NAMI-A towards MCF-7 cells under the same experimental conditions [130,131]. Additionally, the nature of ligands can become critical for interactions with biomolecules at the cellular level. Cellular uptake efficacy and antiproliferative effects exhibited by anticancer drugs are in fact strictly correlated with their hydrophobicity. 

To exploit the advantages of nanotechnology, very recently many ruthenium compounds have been successfully converted into nanomaterials [87,90,91,129,130,131]. Due to both their passive and active tumor targeting properties, diverse nanostructured materials could hire a central role in cancer therapeutics. Indeed, various drawbacks have been observed for non-nanostructured Ru-based materials, principally caused by low stability in the biological environment, ultimately affecting their pharmacokinetic and pharmacodynamic profiles. Besides significantly limiting cellular uptake process, instability/degradation can largely reduce their therapeutic efficacy [121,122]. Because of exchange of chlorido ligands with hydroxyl ions—causing the formation of insoluble poly-oxo species-Ru(III) complexes, i.e., NAMI-A, AziRu and other derivatives, have shown a short half-life in aqueous solution [130,132,133]. Therefore, the design of stable and long-life Ru-based antineoplastic agents is a current challenge to surmount these limitations; in this context, variously decorated nanostructured materials enclosing Ru-based complexes could become an efficient way to administer anticancer drugs in vivo. On this point research is active and several recent reports are showing the antitumor efficacy in preclinical studies of nanostructured materials functionalized with Ru complexes.

## 7. The Idea of Biocompatible Ru(III)-Based Nucleolipid Nanosystems

Moving towards the design of new nanomaterials for biomedical applications in order to develop more effective ruthenium-based treatments and to limit their environmental degradation, about ten years ago we started the design of molecular platforms for the safe delivery of ruthenium complexes in cells. Engaged in this challenge, our group have proposed an innovative approach for Ru(III)-based drugs transport in vivo. By means of nanobiotechnological tools, we have focused on the synthesis and characterization of novel amphiphilic derivatives of nucleosides, i.e., hybrid molecules endowed with lipid moieties linked on the ribose scaffold belonging to the class of nucleolipids [134]. Besides being naturally present in both eukaryotic and prokaryotic cells, wherein they are involved in metabolic processes, nucleolipids have also demonstrated various bioactivities, mostly of antimicrobial, antifungal, antiviral and antitumor types [135]. In this direction, several researchers have attempted hitherto to exploit active nucleolipids in therapeutic area [136]. Their amphiphilic properties make nucleolipids very attractive compounds, able to produce supramolecular structures by spontaneous self-assembly, such as liposomes or multilamellar layers, micelles and/or vesicles, monolayer films, in which normally their nucleobases are exposed to the aqueous phase [137]. Therefore, for their original features, these molecules have been designated as basic scaffolds linked to the AziRu complex. Then, self-assembly into stable nanostructures ensures the efficient transport of the metal into cancer cells. In details, by using ribo- and deoxyribonucleosides—thymidine or uridine—as starting polyfunctional building blocks, a suite of nucleolipidic Ru(III) complexes functionalized by means of different hydrophilic and lipophilic chains and enclosing the AziRu complex has been prepared. The introduction of a pyridine-methyl arm on the nucleobase as a functional ligand allows for the formation of an octaedric complex with the Ru(III) ion. The formulated nucleolipid Ru(III) complexes, named ToThyRu, HoThyRu, and DoHuRu, are characterized by a natural tendency for self-aggregation in physiological conditions (Figure 2). Decoration with one or two aliphatic chains ensures in aqueous solutions the assembly into ordered nanosystems, together with one oligoethylene glycol chain of variable length, acting as a protective “stealth” agent for the resulting nanoaggregates [138].

In 2012, the synthesized molecules have been investigated as pure aggregates, as well as in combination with the naturally occurring lipid palmitoyl-2-oleoyl-sn-glycero-3-phosphocholine (POPC), at selected POPC/Ru molar ratios (85:15) [129]. Indeed, on the outlook for biomimetic nanosystems, the combination of nucleolipid Ru complexes with natural occurring phospholipids allowed us to establish a priori the metal amount to be administered. In addition, the liposome bilayer ensures protection from degradation to the enclosed ruthenium complex (Figure 3). Shortly after, in order to enhance the antineoplastic activity by both increasing the ruthenium content within the aggregates and improving their cell uptake, the coaggregation of the nucleolipids with the cationic lipid 1,2-dioleyl-3- trimethylammonium propane chloride (DOTAP) has been tested. In the latter case, DOTAP/Ru molar ratio can be increased up to 50:50, thus ensuring high metal content to be enclosed within nanoaggregates [130]. Moreover, in depth microstructural investigations, performed by combined approach including different physico-chemical techniques, allowed their stability, size, and shape to be determined. What is remarkable is that, compared to the nude complex AziRu, both zwitterionic and cationic Ru(III)-containing liposomes proved to be stable for diverse months under physiological conditions, validating our innovative strategy based on liposomal nanoaggregates to develop promising lead compounds for forthcoming in vivo studies [129,130]. One of the major drawbacks of low molecular weight Ru complexes was in fact their limited stability. In a biological environment, the degradation process for NAMI-A, AziRu and their derivatives is attributed to the substitution of chloride ions, as well as of the DMSO ligand, with water molecules and hydroxide ions, followed by the formation of poly-oxo species in a relatively short time. This culminates in a detectable color change of the solution couple to brown particles precipitation. Therefore, the development of stable Ru(III) nanoformulations is currently a central requirement in the design of effective Ru-based antineoplastic agents. 

As an evolution of our studies and with the idea to model highly biomimetic liposomes capable of preserving the bioactive ruthenium complex core, we have also proposed a novel cholesterol-containing nucleolipid ruthenium complex, named ToThyCholRu (Figure 2). As before, this system has been studied as such and in parallel when lodged in the biomimetic membrane constituted by POPC [139]. By exploiting the high affinity of cholesterol with natural lipidic molecules, ToThyCholRu has been designed in order to achieve easy penetration through the cell membrane, thus facilitating ruthenium complex internalization. Insertion of one oligo-(ethylene/glycol) chain and one cholesterol residue on the ribosidic sugar ring moiety confers the desired amphiphilic behavior to this nucleolipid. Moreover, the cholesterol moiety favors the insertion of the polyfunctional nucleolipid within the bilayer of POPC liposome thus protecting the Ru complex from possible degradation (Figure 3). The final nanoformulation enclosing AziRu up to 15 mol % is stable for at least several weeks allowing easy handling for biomedical applications [139]. Aiming at further developments of these nanosystems for novel tools for anticancer therapies and following the same pattern, the amphiphilic cholesterol-containing ruthenium complex ToThyCholRu has been next inserted into DOTAP-based cationic liposomes, which contain 30% in moles of AziRu in a stable formulation for several months [140].

Then, in the wake of the promising outcome obtained for the first generation of ruthenium(III) complexes we have just described, and to expand the family of nucleolipid Ru(III) complexes as anticancer drugs, we have developed novel analogs endowed with diverse decorations of the nucleolipid skeleton. With respect to the first series of compounds we have designed (ToThyRu, HoThyRu, and DoHuRu), the pyridine ligand was not attached at the N-3 but at the C-3′ position on the pyrimidine moiety, giving rise to a model compound for a second generation of metal-complexed uridine-based nucleolipids. This compound, named HoUrRu, has an oleic acid residue at the 2′-position and a heptadic (ethylene glycol) chain at the 5′-position (Figure 2). Additionally, in this way the resulting nucleolipid can potentially interact with cellular nucleic acids via Watson–Crick hydrogen bonds or stacking contacts. HoUrRu can be incorporated in both zwitterionic and cationic liposome formulations without significant perturbations in the nanoaggregate morphology. Indeed, with respect to the other amphiphilic Ru-complexes, HoUrRu has showed improved potential to co-aggregate with different lipids and, once lodged in POPC or DOTAP liposomes, no degradation processes were observed in biological microenvironment, with both formulations virtually stable for months [141].

## 8. Preclinical Validation of Nucleolipid Ru-Based Nanoformulations in Models In Vitro

Almost ten years of preclinical studies by selected panels of human cancer models in vitro, and of their healthy counterparts (when available), have allowed us to outline the bioactivity profile of nucleolipid Ru-based nanosystems, to establish the most sensitive tumors to action in vitro of AziRu, as well as to explore the mechanisms underlying the observed antiproliferative effects [87,138]. Thus, based on their efficacy and safety in preclinical test, we have selected the most promising nanoformulations for in vivo studies and for future further developments, such as the possibility of specific functionalizations to achieve an active targeting of cancer cells. In this context, starting from the AS1411 aptamer, which has attracted conspicuous interest as effective ligand for tumor-selective delivery and first to be tested in human clinical trials, we have also investigated a set of its derivatives with different lipophilic tails, ad hoc designed to decorate or to be incorporated by hydrophobic interactions in our liposomal formulations [142,143]. Preliminary results have shown a great deal of promise for future optimization of suitable nanoplatforms, thus assembling a variety of finely tuneable AS1411-decorated supramolecular compounds for effective nanotechnological applications [143].

In line with studies on nucleolipid molecules endowed with biological activity, targeted bioscreen in models in vitro have first allowed us to determine the biocompatibility of nucleolipidic nanosystems before their use as nanovectors [130,137]. Afterward, tested on a large panel of human cancer (breast, colorectal, prostate, gliomas, neuroblastomas, and cervical cancers) and non-cancer cells (non-tumorigenic epithelial cells, keratinocytes, fibroblasts, and macrophages), roughly all the studied Ru(III) nanosystems showed a remarkable anticancer activity in vitro, significantly higher than that of the inspiration molecule NAMI-A, coupled to low cytotoxic profiles in healthy cell models [87]. More in details, the evaluation of a “cell survival index” derived from concentration/effect curves suggested POPC-based nanocarriers as capable of generating cytotoxic effects similar to that of AziRu alone, but at a ruthenium concentration of about 6 times smaller since they enclose only 15% of ruthenium in moles [129]. Through results standardization to the actual ruthenium amounts delivered by nanoaggregates, we concluded these nanocarriers as more active in reducing cell cancer growth than the “naked” AziRu complex. The higher antiproliferative activity was observed in BCC, the most responsive to the ruthenium action in our model, with the IC_50_ values showing an overall decreasing trend along the series NAMI-A, AziRu, DoHuRu/POPC (e.g., 620, 305, and 18.9 µM, respectively, in MCF-7 cells), highlighting the effectiveness of our nanoformulations in ruthenium delivery [129].

As reported above, to further enhance the anticancer activity of amphiphilic ruthenium complexes, we have subsequently developed a novel drug delivery strategy by enhancing the ruthenium content within the nanoaggregates and promoting their cell uptake. This outcome was accomplished through the coaggregation of the amphiphilic ruthenium complexes (having a negative charge) with the cationic lipid DOTAP, obtaining nanoaggregates specifically designed to exhibit high stability in biological microenvironment even at high Ru-complex content (50:50 in moles) [130]. Our results showed that all the nanoaggregates of HoThyRu, DoHuRu, ToThyRu, and HoUrRu mixed with DOTAP are particularly efficient. Indeed, as underscored by the potentiating factors, IC_50_ values for these ruthenium complexes are in the low micromolar range (around 10 μM) on MCF-7 cells, proving to be more effective than AziRu (305 μM) assessed in the same experimental conditions [130]. Throughout our project, this was a central outcome validating the efficacy toward cancer cells of nucleolipid Ru(III) complexes lodged in cationic DOTAP liposomes. We supposed these results as being coupled to the cationic nature of DOTAP nanoformulations, that can favor interactions with the cell membranes for drug cellular uptake. As shown in Table 1, Ru(III) derivatives under evaluation were more active against various subtypes of BCC, though some antiproliferative effects observed toward malignant cells of different histogenesis are also worthy of note, for example in prostate and colorectal cancer (ruthenium IC_50_ of 7.7 and 12 µM, respectively, by using ToThyRu/DOTAP and HoUrRu/DOTAP nanosystems) [130]. By anticancer activity screening performed on tumor cell panels throughout preclinical investigations, we have in fact proven that nucleolipid Ru-containing nanosystems exhibit a constant and remarkable activity against all the BC lines tested, including TNBC, while their effects are more variable in cancer cells other than BC, where they show more isolated positive results [87,138]. Moreover, in some subtypes of BC cells, such as MCF-7, but especially in their variants CG-5 and in T47D, we have detected the most remarkable antiproliferative activities (IC_50_ less than 5 µM) by means of the DOTAP-based cationic nanosystems [57,78]. 

In addition to other prospective intracellular molecular targets, metal-based anticancer drugs target predominantly DNA in the nuclear cage, so that the uptake kinetic has become as a central element affecting their antiproliferative efficacy. For both ruthenium and platinum complexes, the cytotoxicity profiles are in fact closely associated with the ratio of cellular uptake, which in turn is positively coupled to their lipophilia, e.g., AziRu is more cytotoxic than NAMI-A because of higher lipophilicity ensuring improved cellular uptake [129,130,144]. Time-course experiments by confocal microscopy and ad hoc designed fluorescently tagged analogs of DOTAP-based nanoformulations proved they have high propensity to cross cell membranes and to accumulate in a fast process into cells (Figure 4) [78]. Indeed, due to their both amphiphilic features and positive net superficial charge, and thereby starting from charge attraction and tight contact with the target membrane, these cationic nucleolipid Ru nanoaggregates can easily cross cell membranes via either fusion or endocytosis by nonspecific molecular patterns. In accordance, sub-cellular bioaccumulation and localization of the Ru(III) complex evaluated by inductively coupled plasma-mass spectrometry (ICP-MS) analysis next to DOTAP liposomes application to MCF-7 cells, further support that cellular internalization is significantly boosted by the nanoformulation compared to the “naked” AziRu complex. Nano-delivered ruthenium was extensively distributed in intracellular compartments, but especially in nuclei wherein it was found in DNA (Figure 4) [78]. Behind a positive feedback on cellular entry, fluorescent patterns and more in general these outcomes suggested a liposome disaggregation/degradation combined with the subsequent spread of the active metal agent in different compartments of cancer cells [78,129,130].

## 9. Selectivity and Efficacy of Nucleolipid Ru-Based Nanoformulations in BCC Models

Along this path, both neutral and cationic liposomes enclosing the nucleolipid Ru(III) complexes have exhibited selectivity towards specific cancer types. Outcome from estrogen and progesterone receptor positive adenocarcinoma cells (e.g., MCF-7 and CG-5 cells) were of special interest [57,78]. Indeed, the most remarkable cytotoxic effects throughout our project have been detected in ER BC models in vitro, which roughly reflect the characteristics of luminal BCC in vivo, a widespread aggressive tumor in women [87,129,130,138,139,140,141,145]. Interestingly, these findings are consistent with many reports showing for ruthenium complexes significant antiproliferative effects against specific subtypes of BC [124]. Thus, starting from these data, to expand knowledge for an optimal Ru-based candidate drugs development as chemotherapeutic options, we have next focused on selected in vitro models of BC—i.e., MCF-7, CG-5, MDA-MB-231, MDA-MB-468, and MDA-MB-436 cell lines—with distinct phenotypic and/or genotypic features, and different replicative and/or invasive potential. At preclinical level, BC is modelled by established experimental cell lines, whose features are critical in translational BC research. Nowadays, the ER adenocarcinoma MCF-7 and the TN breast adenocarcinoma MDA-MB-231 cells are believed as the most consistent models in vitro of BCs [146]. Using these models, cationic Ru/DOTAP nanoformulations have always emerged as the most effective in reducing the proliferation of BCC. In this context, given that cisplatin (*c*DDP) is used in guideline treatment protocols for various BCs, and that these BCC are sensitive to cisplatin *in vitro*, we used *c*DDP as cytotoxic reference drug [33,82]. Results standardization in favor of the ruthenium aliquot included in DOTAP nanosystems reveals IC_50_ values of about 3÷4 μM in CG-5 cells, and of about 10 μM in MCF-7 cells, indicative of a remarkable antiproliferative bioactivity. In sum, ruthenium IC_50_ values for DOTAP nanoformulations are placed in the low micromolar range (generally less than 20 μM), comparable or even lower than IC_50_ measured for *c*DDP under identical experimental conditions (see Table 1) [57,78]. Ruthenium IC_50_ values for POPC nanoformulations are also relevant, reaching values below 20 μM as in the case of MDA-MB-468, MCF-7 and CG-5 cancer cells [57]. In the same BCC models, the “naked” AziRu complex shows milder antiproliferative effects, associated with higher IC_50_ values (>250 μM) [57]. Thus, the potentiating factors for Ru(III)-containing liposomes with respect to AziRu show values higher than 20 as antiproliferative ability [130]. IC_50_ values for AziRu are mostly in line with those described for NAMI-A. An appropriate delivery strategy is thereby critical to guarantee drug stability in the extra-cellular environment, as well as transport across membranes and the bioavailability at the biological targets. Notable, throughout preclinical studies the cationic Ru-based nanosystems have proven to be essentially inactive on MCF-10A cells, employed as a consistent model in vitro for normal human mammary epithelial, thereby supporting their selectivity of action towards malignant cell models of BC [78]. All the main statistical dataset concerning in vitro evaluations has been included in a short section as Appendix A.

Hence, preclinical evaluations provided convincing and definitive evidence validating our nucleolipid Ru(III) nanosystems as promising options among advanced therapeutics with superior anticancer activity.

## 10. Biological Responses to Nucleolipid Ru-Based Nanoformulations in BCC Models

Based on a growing morbidity and high mortality rate of cancer, there is an ever-increasing demand for the development of novel therapeutic compounds. Present-day research activities attempt to give a deeper knowledge on cellular responses, including chemoresistance, to anticancer treatments, wherein the activation of cell death pathways, i.e., apoptosis and autophagy, can play a central role in specific cancer types [46,51,54,64,67]. Though its role is being decreased by the advancement in targeted therapies, to date chemotherapy still occupies a crucial position in clinic for cancer therapy [147,148]. In this context, drugs acting on multiple targets can limit chemoresistance and preserve their efficacy, being thus considered as the upcoming agents for innovative anticancer interventions [149]. Moreover, concerning BC no targeted drug has so far been approved for the therapy of the most aggressive TN subtype. Accordingly, we have demonstrated that amphiphilic ruthenium complexes are capable to inhibit BCC by stimulating specific programmed cell death pathways, sometimes combined to cellular autophagy [57,78,87,138]. To reduce chemoresistance, as well as to fight uncontrolled proliferation, the activation of multiple death pathways via potential interactions with both nuclear and cytosolic targets by chemotherapeutics is a largely desired objective, especially in aggressive cancer diseases such as TNBC with limited treatment options [36,150]. Tissue homeostasis in multicellular organisms is regulated by balancing signals which promote proliferation or removal of cells via apoptosis. As well, normal mammary gland homeostasis is regulated by a balancing among growth and apoptosis [151]. In the context of cancer, genetic anomalies and unconventional microenvironments profit by endogenous pro-survival signaling to re-program and rewire metabolism to sustain survival, growth, and proliferation. It is now largely established that tumor is not simply a consequence of deregulated proliferation, but also of evasion from apoptosis, so the regulation of programmed cell death in cancer cells can be of critical importance in defining the overall growth or regression in response to treatments [45,47,152]. Besides being a major occurrence in the onset and progression of BC, malignant cells survival represents a crucial contributing factor also in clinical failure and chemoresistance development [153]. Therefore, metabolic alterations in cancer contributing to apoptosis escape and therapy resistance have become a central issue to be elucidated. Several distinct ruthenium compounds have been shown to exert their antitumor activities by various modes of action, so it was not possible to unambiguously define a precise mechanism at the molecular level [98]. In analogy with *c*DDP - having nuclear DNA as final target with resultant adducts formation and cell cycle arrest - it has been proved that also AziRu can interact with DNA models, wherein Ru(III) ions can bind stably nucleotide structures [154,155]. Nevertheless, we have broadly discussed about the possibility for both Ru(II) and (III)-based drugs to interfere with both intra- and extra-cellular protein targets. In this frame, new interest has been recently placed on metal-based drugs that can inhibit enzymatic activities and target proteins directly. These interactions could be of crucial importance to enlighten the antiproliferative effects reported on this topic by several groups [156]. In addition, the occurrence of distinct hallmarks of apoptosis due to protein-targeting after ruthenium complexes administration has been extensively stressed; consistently, in preclinical investigations we have demonstrated an invariable stimulation of the apoptotic machinery [57,58]. Thus, we can now assume that the primary mode of action in BC models of nucleolipid AziRu complexes lodged in both POPC and DOTAP nanoformulations is the activation of apoptosis (see Table 2) [87].

Notwithstanding, for the most effective DOTAP formulations, the concurrent induction of autophagy can play an important role in defining the overall anticancer effect, representing a prospective further possibility of interfering with BCC uncontrolled proliferation [78,87]. As discussed before, the cell uptake kinetics observed with the use of cationic nanoformulations could be somehow engaged in promoting the simultaneous induction of diverse cell death mechanisms. Considerable quantities of the Ru(III) complex enter in fact in cancer cell, having thereby the possibility to interact with a variety of molecular targets [78]. For what concerns apoptosis, cytomorphological analysis and DNA fragmentation supplied further data of an apoptosis-inducing activity. All the formulations containing nucleolipidic Ru(III)-complexes can trigger the mitochondrial apoptotic cell death pathway in BBC via activation of caspase-9, wherein pro-apoptotic cell death members of the Bcl-2 family heighten mitochondrial permeability sustaining the whole process normally triggered by stress signals. Bax and Bak seem to be the core regulators for the intrinsic pathway commitment. Upon apoptotic stimuli, these and other factors are recruited for oligomerization at the mitochondrial outer membrane (MOM) to modulate its permeabilization, which is considered a major step in apoptosis [157]. Interestingly, in BC preclinical models this occurs independently of the effector caspase-3, as proven by the restoration of mitochondrial apoptosis in MCF-7 cells, probably unresponsive to some chemotherapeutics for an inherited lack of caspase-3 [57,158]. Ongoing studies on a repertoire of candidate drugs have demonstrated that apoptotic defects in cancer could be bypassed by therapeutic interventions directly impairing the mitochondrial function, and capable of restoring the regulation of programmed cell death [159]. The pharmacologic renewal of apoptosis, either by inhibiting antiapoptotic proteins/regulators or by inducing pro-apoptotic factors, retains solid potential for cancer interventions [159]. Correspondingly, AziRu may likewise act at the mitochondrial level on specific targets. Among Bcl-2 family, the oncogene Bcl-2 is a key anti-apoptotic factor boosting cellular survival, while Bax, as a pro-apoptotic effector, promotes mitochondrial membrane permeabilization as central signal in the apoptotic flux [54]. Many clinical trials support the occurrence of an over-expression of the pro-survival Bcl-2 protein, including a down-regulation of the pro-apoptotic Bax, as negative prognostic marker in patients with breast and other cancers [46]. Very interestingly, in BBC, both ER and TN subtypes, the substantial rise in the Bax/Bcl-2 ratio we have demonstrated following ruthenium treatment by POPC and DOTAP nanoformulations is likely associated to the activation of intrinsic apoptosis [57]. This suggests for AziRu the possible occurrence of mitochondrial molecular targets capable of restoring the switch of cancer cell death. In accordance, many tumors are responsive to Bcl-2 regulation as key inducer of proliferation, as well as increased Bax levels have been coupled to an enhanced response to chemotherapy [40,47,50]. At the mitochondrial level, Venetoclax is the first FDA-approved drug in cancer directly targeting the anti-apoptotic Bcl-2 protein to reactivate apoptosis. By an “inhibition of the inhibitor”, the drug deactivates its ability to counteract pro-apoptotic factors including Bax [160]. Thus, activation of pro-apoptotic factors by direct or indirect targeting can provide blueprints for the design of apoptosis regulators to interrupt pathologic cell survival.

Noteworthy, DOTAP formulations can concurrently activate the extrinsic apoptotic pathway in TNBC, thus offering an additional possibility to block the growth of this BC type. Indeed, MDA-MB-231 cells, in which the intrinsic apoptosis is by now detectable following incubation with DOTAP-based nanosystems, reveal an evident cleavage of pro-caspase-8 to provide active fragments involved in the activation of the extrinsic pathway (see Table 2) [57]. We have supposed that the cationic DOTAP nanoaggregates, owing to their chemical nature and surface charge, can establish closely contact with the cell membranes [130,140]. Meanwhile, specific molecular contacts with membrane components plus possible drug release might stimulate specific surface receptors involved in the early signaling of extrinsic apoptosis, as proven by some Ru-based chemotherapeutics whose action has been correlated to initiation of this apoptotic pathways [58,161]. In this context, the clarification of the interaction between metallotherapeutics and cell membrane may offer further valuable assumption for rational design of metal-based anticancer drugs.

In addition to apoptosis, DOTAP nanoformulations enclosing the nucleolipid AziRu complex trigger extensive and massive autophagy activation in BCC models, including ER and TN subtypes, providing a crucial drive in causing cell death in vitro [78]. Indeed, sustained induction of autophagy may hamper cancer cell survival acting as a tumor suppressor factor, so that a number of anticancer therapeutic agents as inducers of the autophagic flux are under development [64,67,162,163]. Moreover, a few recent evidences support the occurrence of apoptosis together with autophagy in cancer as a consequence of precise signaling, probably involving drug-dependent dysfunction of mitochondria. Researches underscore the crucial connection amongst apoptosis and autophagy, and propose apoptosis as frequently associated to enhanced autophagy probably via interference with Bcl-2 family members [81]. Likewise, cancer cells with enhanced autophagy show a less aggressive behavior, including a greater responsiveness to chemotherapy [164,165]. Thus, despite the complex dual role of autophagy either as tumor suppressor or cell survival factor, depending on several dynamics, autophagic cell death as therapeutic approach in anticancer interventions is nowadays the topic of several investigations [65]. In this framework, some Ru(II) complexes have already shown to activate autophagy in cancer cells as a cell death mechanism [79,80,166]. In line, in BBC models our cationic Ru(III) nanosystems induce a significant rise in both forms of LC3 protein, wherein the autophagosomal marker LC3-II (phosphatidylethanolamine conjugate) reflects to a large extent the autophagic activity. Moreover, under the same experimental conditions, we reported that Beclin 1 expression—the first recognized Bcl-2 interacting mammalian gene regulating autophagy among Atg proteins—was positively correlated with the autophagic process after treatments in BCC models (see Table 2) [57,78]. In support of these findings, a few recent data show autophagic machinery defects next to Beclin 1 lack under different circumstances, as well as chemotherapeutic interventions. Moreover, Beclin 1 is often reduced in BCs including TNBC, and this could have adverse effects on cancer growth [167,168]. Indeed, the inducer regulatory role of Beclin 1 in the autophagic flux in MCF-7 cells has been linked with cellular proliferation slowdown [169]. Therefore, given that excessive autophagy is basically incompatible with cell survival, its sustained stimulation by specific therapeutic interventions, as an additional exploitable mechanism of cell death, claims new studies and further insights. As ruthenotherapy additionally activates apoptosis in BCC, common upstream signals may be activated, resulting in combined types of different cell death. A molecular model is proposed in Figure 5. Latest studies have in fact suggested interactions between Bcl-2 proteins and Beclin 1, including Atg members other than Beclin 1, as conceivable network in autophagic and apoptotic crosstalk [170]. Just as a blueprint, since the oncogenic Bcl-2 hinders both apoptosis and autophagy by Bax and Beclin 1 inhibition, its decreased expression in response to “ruthenotherapy” could be somehow involved in re-activation of mechanisms of cell death in BCC. However, assumed the potential for upcoming advanced cancer treatments, crosstalk among apoptosis and autophagy are still controversial and remains to be defined to a large extent, as well as in specific subtypes of BCC. Their elucidation could open new perspectives for a final breakthrough on Ru(III)-based nanosystems as effective multitarget anticancer weapons in refractory cancers such as TNBC.

## 11. Conclusive Remarks and Future Perspectives

Breast cancer is certainly among the most studied disease afflicting humankind. This is not surprising given the overall number of women diagnosed with the malignancy, making it a main concern worldwide. In recent years, researchers have made great progress in the management of BC. Nevertheless, mainly because of molecular subtypes heterogeneity and chemoresistance to treatments, the battle is still far from being won and must be fought on several fronts. Although the most exciting area of research are perchance immunotherapy and targeted therapy, chemotherapy is still a necessary weapon to cope with invasive and widespread tumors such as TNBC. Among chemotherapeutics, metal-based drugs have dominated the scene so far, with the research that has increasingly moved on other precious metal alternative to platinum. Many scientists have focused their interest on the design of original ruthenium complexes due to their prospective biomedical and pharmaceutical appliance. Indeed, ruthenium exhibits many ideal drug properties, making it a highly favorable therapeutic agent, including its redox abilities, minimal toxicity profile for healthy cells, high selectivity for cancer cells, ligand exchange properties, and favorable adsorption properties. In this framework, nanostructured materials functionalized with Ru complexes have proved to be a useful and safe way to administer Ru-based drugs for cancer treatment. Herein, starting from ‘traditional’ Ru-based compounds, we have highlighted the recent developments in a specific type of nucleolipid nanosystems enclosing the anticancer ruthenium(III) complex AziRu, including extensive information about the possibility of their decoration and functionalization for a selective and effective drug delivery to tumors. With the aim of expanding the beneficial properties of available amphiphilic Ru(III) complexes, we are investigating novel biocompatible platforms alternative to nucleosides to develop new prospective anticancer agents. Very recently, a trifunctional α-amino acid, in place of the nucleosidic core of amphiphilic Ru(III) complexes, has been selected. The amino acidic scaffold, coupled to the Ru(III) complex, is decorated with both hydrophilic and lipophilic moieties, conferring high tendency to form stable aggregates which are showing promising anticancer activity [171]. Moreover, with the aim of designing increasingly biocompatible liposomal formulations, we are also following an attractive path based on the use of natural amphiphilic molecules such as lipooligosaccharides (LOS) from Gram-negative bacteria. Besides displaying interesting biological properties, when inserted into nucleolipid Ru(III)-based nanoformulations specific LOS have very poor inflammatory properties and a core region that should impart stealth properties to the liposomes, possibly ensuring a long time in circulation [172].

Beyond chemotherapeutic interventions by nanostructured material, as part of a potential combined targeted therapies planned to improve the anticancer efficacy of Ru-based strategy, and, in the meantime, to overcome the possible occurrence of chemoresistance, a further approach we are pursuing encompasses interference with multiple intracellular oncogenic pathways: in concert with the uncontrolled growth mechanisms counteracted by ruthenotherapy, the simultaneous intercepting of aberrant miRNA expression can be experimented in TNBC patients through the employ of ad hoc designed cationic nucleolipid nanoformulations loaded with both the ruthenium(III) complex AziRu and a selected miRNA with proven antitumor activity towards human BC [173,174].

Overall, in this review we have attempted to showcase the path toward the preclinical validation of novel amphiphilic nanomaterials designed for effective Ru-based anticancer treatments, to be placed within the current metallodrug approach leading over the past decade to advanced multitarget agents endowed with limited toxicity and resistance. On this point, modern chemotherapeutics must in fact address resistance issues that are afflicting many therapeutic regimens. Besides potent drug combinations, chemoresistance can be overcome by using a multitarget approach to improve anticancer efficacy. Indeed, simultaneous action on multiple molecular targets can increase the chances of reactivating death pathways that are commonly suppressed in cancer cells by the action of oncogenes. To our knowledge, ruthenium complexes show a higher activity in several resistant cell lines, including BCC [124,175]. In this direction, the Ru-based nanosystems described herein not only can target DNA, but also enzymes and cellular proteins (Figure 5). Indeed, evidence suggests, as an alternative, proteic molecular targets with which ruthenium can interact. Therefore, although many pharmacodynamic aspects need to be clarified, the landmark remains that of multitarget compounds capable of interacting with different cellular biomolecules. In this way Ru complexes can also activate DNA-independent apoptosis. Potential interactions with varied cellular biomolecules are thereby the most important difference with respect to platinum-based drugs, targeting exclusively nuclear DNA as alkylating agents. In this context, the ability of ruthenium to replace iron in some metalloproteins should not be overlooked [91]. Concerning our Ru(III) containing nanosystems, as they are in preclinical evaluation, it is still difficult to discover intrinsic or acquired resistance phenomena; based on preclinical bioscreens, the observed bioactivities at the moment would not support the occurrence of significant resistance mechanisms. Therefore, this approach may represent a viable option mostly for TNBC treatment, wherein it has not been conceivable to develop targeted therapies so far. Indeed, a breakthrough for clinical appliance of this type of chemotherapeutics requires the precise identification of target biomolecules, as well as the elucidation of the interplay between the death mechanisms triggered either by membrane and/or intracellular interactions. Consideration should be given to the chemical variety of ruthenium complexes being tested, as well as the possibility of insertion into different nanosystems with original properties, which could result in distinct modes of action. However, moving in this direction some initial structure-activity relationships could be established, together with molecular modelling studies to characterize the binding behaviour for a further rational upgrading of advanced metal-based drugs in the perspective of a forthcoming “ruthenotherapy”.

## Figures and Tables

**Figure 1 cells-09-01412-f001:**
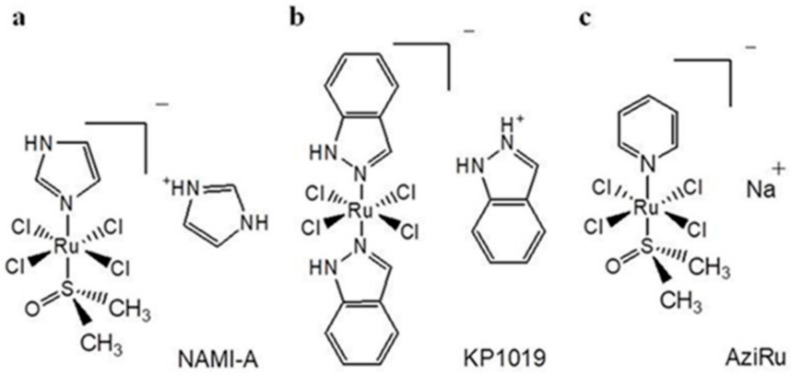
Molecular structures of the Ru(III) complexes NAMI-A (**a**), KP1019 (**b**) and AziRu (**c**).

**Figure 2 cells-09-01412-f002:**
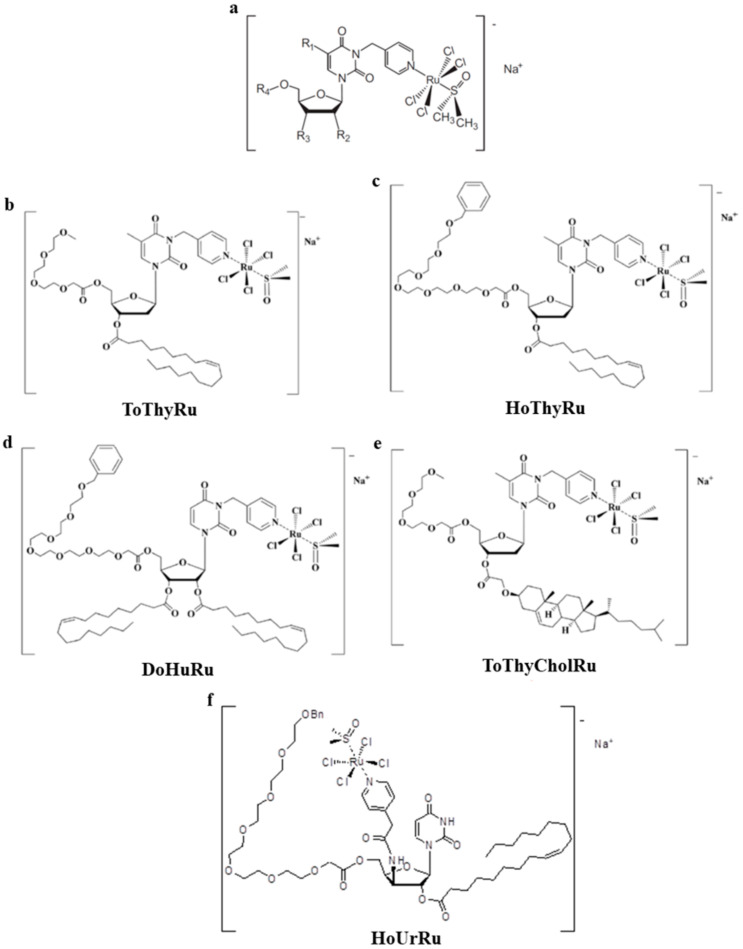
Molecular structures of generic thymidine- or uridine-based nucleolipid Ru(III) complexes (**a**), and of the functional nucleolipid Ru(III) complexes ToThyRu (**b**), HoThyRu (**c**), DoHuRu (**d**), and ToThyCholRu (**e**). HoUrRu (**f**) is a lead compound for a second generation of metal-complexed uridine-based nucleolipids.

**Figure 3 cells-09-01412-f003:**
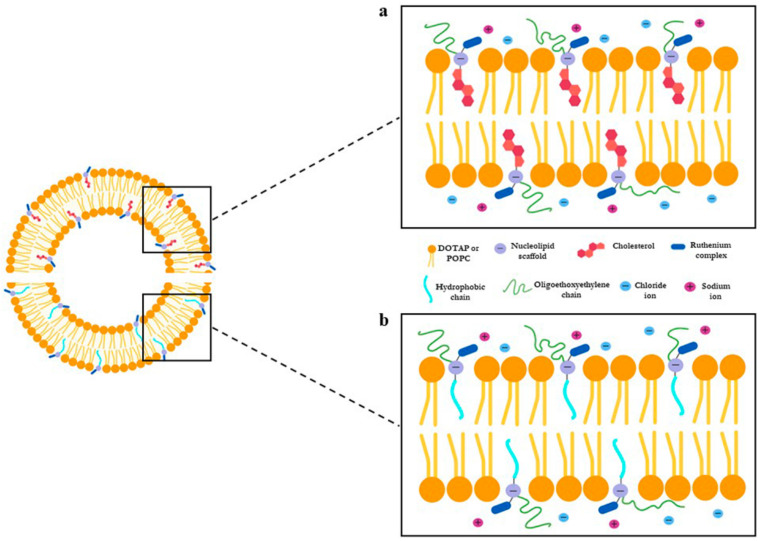
Qualitative molecular representation of the nucleolipid Ru(III) complexes ToThyCholRu (**a**) and DoHuRu (**b**), lodged in POPC (zwitterionic) or DOTAP (cationic) liposome bilayers, as indicated [129,130,139,140].

**Figure 4 cells-09-01412-f004:**
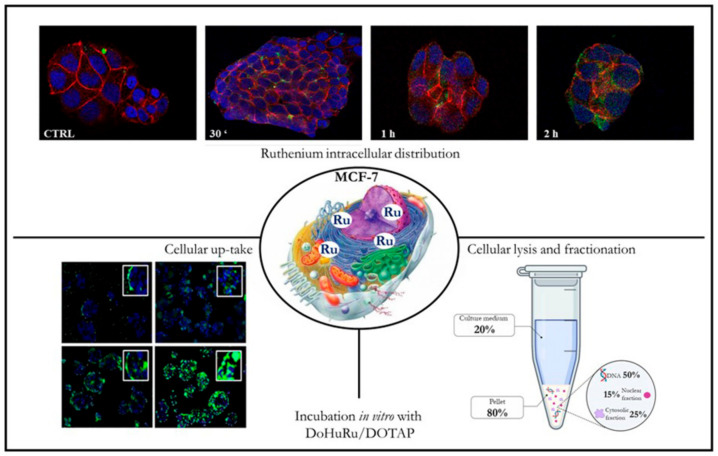
Uptake, distribution, and bioaccumulation of the Ru(III) complex AziRu after incubation of MCF-7 cells with the cationic nanosystem DoHuRu/DOTAP. Cellular uptake and distribution were monitored by confocal microscopy and ad hoc designed fluorescently tagged analogs of DOTAP-based nanoformulations [78,130]. Localization and bioaccumulation of the Ru(III) complex were evaluated in MCF-7 cells and culture media by means of subcellular fractionation and inductively coupled plasma-mass spectrometry (ICP-MS) [78]. In the indicated fractions, ruthenium amounts are reported as percentage of the total ruthenium administered during experiments.

**Figure 5 cells-09-01412-f005:**
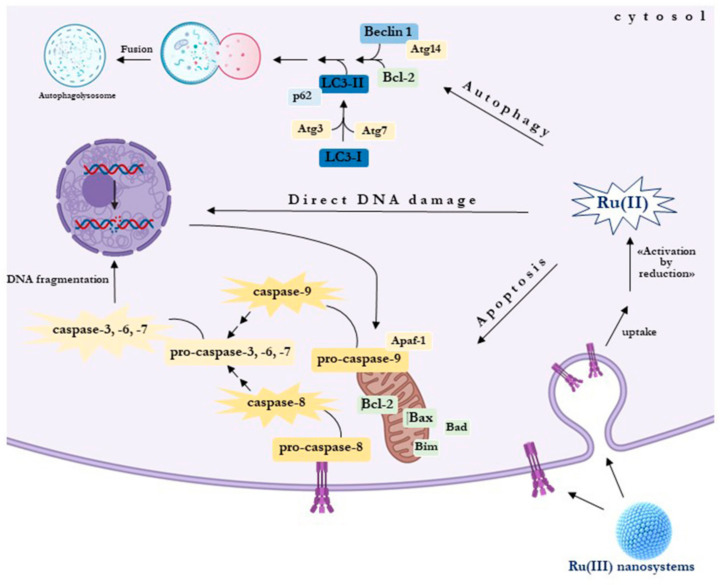
Molecular model proposed to describe the mechanism of action underlying the antiproliferative effect in BBC of nucleolipid nanosystems delivering the Ru(III) complex AziRu. Following uptake and selective activation *in situ*, this model highlights the multitarget action of the drug on the activation of both apoptotic and autophagic cell death pathways.

**Table 1 cells-09-01412-t001:** Best IC_50_ values (μM) relative to the AziRu complex lodged in POPC and DOTAP liposomes (the specific nucleolipid Ru complex is indicated in brackets) measured in preclinical models, including cancers other than BC [57,87]. IC_50_ values for cisplatin (*c*DDP), as cytotoxic reference drug, are included for comparison. Noteworthy, in the same models *in vitro*, IC_50_ values for the naked AziRu complex are constantly higher than 250 µM. (n.a. = not assessed).

IC_50_ (µM)			
	POPC Liposomes	DOTAP Liposomes	*c*DDP
Breast cancer cells			
ER-positive			
MCF-7	18.9 ± 0.1 [DoHuRu]	10.1 ± 0.1 [ToThyRu]	17 ± 5
CG-5	19.4 ± 0.2 [ToThyRu]	3.3 ± 0.2 [DoHuRu]	n.a.
TNBC			
MDA-MB-231	15 ± 1 [DoHuRu]	10.8 ± 0.2 [ToThyRu]	19 ± 4
MDA-MB-436	37 ± 1 [DoHuRu]	15 ± 0.2 [ToThyRu]	n.a.
MDA-MB-468	15.7 ± 0.1 [ToThyRu]	14.2 ± 0.1 [DoHuRu]	24 ± 1
Other cancer cells			
WiDr	20 ± 8 [HoUrRu]	12 ± 5 [HoUrRu]	n.a.
HeLa	25 ± 3 [ToThyCholRu]	34 ± 4 [ToThyCholRu]	10.1 ± 3
LN-229	>75 [ToThyRu]	7.7 ± 1 [ToThyRu]	n.a.
U87-MG	19.8 ± 0.1 [DoHuRu]	11.7 ± 0.1 [ToThyRu]	11.7 ± 0.5
C6	24 ± 5 [DoHuRu]	34 ± 9 [DoHuRu]	6.8 ± 0.3

**Table 2 cells-09-01412-t002:** Main biological effects observed at both nuclear and cytosolic level (with particular reference to mitochondria) throughout preclinical test in ER (MCF-7) and TN (MDA-MB-231) breast cancer (BC) models following incubations with the indicated neutral (POPC) and cationic (DOTAP) nanoformulations [57,78].

Ruthenium Nanosystems	Cell Line	IC_50_ Values (µM)	Main Results
DoHuRu/POPC	MCF-7	18.9 ± 0.1	DNA fragmentation, activation of
			pro-caspase-9, ↑Bax, ↓Bcl-2
DoHuRu/DOTAP	MCF-7	10.3 ± 0.2	DNA fragmentation, activation of
			pro-caspase-9, ↑Bax, ↓Bcl-2,
			↑LC3-I, ↑LC3-II, ↑Beclin 1
DoHuRu/POPC	MDA-MB-231	15.0 ± 1	DNA fragmentation, activation of
			pro-caspase-9, activation of pro-
			caspase-3, ↑Bax, ↓Bcl-2
DoHuRu/DOTAP	MDA-MB-231	12.1 ± 0.3	DNA fragmentation, activation of
			pro-caspase-9, activation of pro-
			caspase-8, activation of pro-caspase-3,
			↑Bax, ↓Bcl-2, ↑LC3-I, ↑LC3-II, ↑Beclin 1

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
