# Peer review of "Breast Cancer Chemotherapeutic Options: A General Overview on the Preclinical Validation of a Multi-Target Ruthenium(III) Complex Lodged in Nucleolipid Nanosystems"

_cells, 2020, doi:10.3390/cells9061412_

Round 1
Reviewer 1 Report
This review paper is well-written and provides a good summary on pre-clinical validation of a multi-target ruthenium complex lodged in nucleolipid nanosystems. Although the paper talks about a pre-clinical study and there may be less statistics involved, it may be useful to include a small section on statistical concerns in general, including a general rule/suggestion on how many biological replicates or technical replicates, how to handle false discovery rate, as well as some good bioinformatics approaches. The main goal is to increase the reproducibility and to get more reliable results based on the most explored and advanced bioinformatic techniques.
Author Response
This review paper is well-written and provides a good summary on pre-clinical validation of a multi-target ruthenium complex lodged in nucleolipid nanosystems
Although the paper talks about a pre-clinical study and there may be less statistics involved, it may be useful to include a small section on statistical concerns in general, including a general rule/suggestion on how many biological replicates or technical replicates, how to handle false discovery rate, as well as some good bioinformatics approaches. The main goal is to increase the reproducibility and to get more reliable results based on the most explored and advanced bioinformatic techniques.
We are grateful to the reviewer for her/his valuable comments and in line with recommendations we have improved the review by including a statistical dataset section as supplementary material which focuses on data and statistical concerns emerged during the experimental activity. A reference to this section has also been included in the main text, in the concluding part of “Selectivity and efficacy of nucleolipid Ru-based nanoformulations in BCC models”, following the discussion of preclinical studies.
Reviewer 2 Report
Ferraro et al. reviewed the positioning of ruthenium-based anticancer drugs in breast cancer treatment, and this reviewer thinks the review is very interesting. However, before accepting this review, I would like the authors to clarify the following points.
1. It seems that the explanation of the mechanism of action of ruthenium anticancer drugs is not sufficient. I think that the most significant difference with platinum drugs seems to be the target molecules, so I would like the authors to clarify the mecahanisms of drug.
2. The authors should describe whether or not there is any information on resistance to ruthenium-based therapy.
3. The authors should describe the difference in adverse events between ruthenium-based therapy and platinum-based therapy. This information must be useful in designing the appropriate therapeutic strategy for the patient with metastatic breast cancer.
Author Response
Ferraro et al. reviewed the positioning of ruthenium-based anticancer drugs in breast cancer treatment, and this reviewer thinks the review is very interesting. However, before accepting this review, I would like the authors to clarify the following points.
It seems that the explanation of the mechanism of action of ruthenium anticancer drugs is not sufficient. I think that the most significant difference with platinum drugs seems to be the target molecules, so I would like the authors to clarify the mechanisms of drug.
We express thanks to the referee for her/his helpful suggestions which have allowed to improve some central aspects of this review. In the wake of these comments, we further stressed the differences between conventional platinum-based chemotherapy and potential new generation ruthenium-based drugs. Obviously, in this frame it must also be considered that, compared to cisplatin in clinical practice since 1978, data currently available for ruthenium complexes are still weak, especially as regards the clinical field. However, the main mechanism of action for some Ru-based drugs is believed to be the activation of cell death pathways such as apoptosis. Nevertheless, much remains to be clarified concerning this issue. In addition, consideration should be given to the chemical variety of ruthenium complexes being tested, as well as the possibility of insertion into different nanosystems with original properties, which could result in distinct modes of action. For sure, although the final effect is the same as that of cisplatin, i.e. apoptosis activation, it is assumed that the molecular targets responsible for the observed effects of Ru containing candidate drugs can be multiple and not necessarily nuclear DNA. Nevertheless, as discussed in the review, fluorescence experiments for cellular uptake and trafficking show that BCC treatment with our Ru(III)-containing nanosystems finally results in large nuclear drug accumulation, just as in the case of cisplatin. Other evidence suggests, as an alternative, proteic molecular targets. Therefore, although many pharmacodynamic aspects need to be clarified, the landmark remains that of multitarget compounds capable of interacting with different cellular biomolecules. Maybe in this way Ru complexes can activate DNA-independent apoptosis. This is the most important difference with respect to platinum-based drugs acting exclusively as alkylating agents. In this context, the ability of ruthenium to replace iron in some proteins should not be overlooked. Therefore, on this topic, we have included an integration in the final section of the review, as well as in the opening one on Ru-based drugs.
The authors should describe whether or not there is any information on resistance to ruthenium-based therapy.
One of the aims of developing novel classes of anticancer ruthenium complexes consists precisely in overcoming chemoresistance, especially in tumor cells unresponsive to conventional treatments, such as platinum-based therapy. Indeed, to limit chemoresistance, multiple cell death pathways induction by novel metal-based chemotherapeutics is believed as a very promising strategy. To our knowledge, ruthenium complexes show a higher activity in several resistant cell lines, including BCC. Concerning our Ru(III) containing nanosystems, as they are in preclinical evaluation, it is still difficult to discover intrinsic or acquired resistance phenomena. However, based on bioscreens in vitro, the observed bioactivities now would not support the occurrence of significant resistance mechanisms. Probably multitarget action, coupled to an effective nanodelivery, plays a major role. However, according to the reviewer suggestion, we have now included some additional sentences about resistance to ruthenium-based therapy in the final section.
The authors should describe the difference in adverse events between ruthenium-based therapy and platinum-based therapy. This information must be useful in designing the appropriate therapeutic strategy for the patient with metastatic breast cancer.
Within the review we by now advise the prospective low toxicity profile of new ruthenium-based candidate drugs, especially compared to the well-known adverse effects of Cisplatin and other conventional therapy regimens. This is another central aspect prompting the development of a new generation of compounds in the field of metallochemotherapeutics. However, roughly like discussed in point 2, as the clinical trials are currently little explored, except for NAMI-A and NKP1339, it is not easy to address adverse effects in the clinic. Certainly, our in vitro data and preliminary in vivo studies seem to support a rather low toxicity profile compared to that of cisplatin. But this will be the topic of an upcoming report by using models in vivo, wherein we start from the profound intrinsic differences between ruthenium-based and conventional platinum-based drugs, being the latter much more toxic by their alkylating nature. Naturally, in this case as well, the role of a selective nanodelivery system can be decisive to limit biological effects on healthy tissues. In addition, as already discussed throughout the manuscript, activation by reduction in situ of Ru(III) complexes can make a significant contribution to reduce adverse effects.
Reviewer 3 Report
Evaluation: The manuscript has a poor organization and is hard to follow.
- Section 2. The topic for this section is “Approved Therapy for BC Treatment.” But the current description in this section is NOT focused on the approved treatments. Corresponding to the subtitle for this section, all current treatments should be listed and when they were approved. Also, the references cited to support this section are not appropriated because most of them are not the original reports.
- Section 3. This section is poorly organized and doesn’t make sense to be listed in this manuscript. It is very hard to follow why controlling apoptosis and cell death/cell survival can be prospective targeted therapies for breast cancer.
- Section 4. This section is very weak as well. The relationship between autophagy in breast cancer cells and “ruthenotherapy” is very weak.
- Line 316, the advantageous properties of ruthenium complexes should be specified here.
- Line 415, specification of the multifactorial concerns (physical, chemical, and biological) would let the readers to easily follow the manuscript.
- Line 424, “both inorganic and organic” should be placed after “derivatives.”
- The data from Table 7 did not show significant selectivity of antiproliferative activity among different cell lines. Consequently, the section 8 doesn’t make sense to me. The authors should explain what kind of selectivity exists. The reasons for the selectivity should be illustrated.
Collectively, this manuscript has a lot of problems which need to be fixed before publication.
Author Response
Section 2. The topic for this section is “Approved Therapy for BC Treatment.” But the current description in this section is NOT focused on the approved treatments. Corresponding to the subtitle for this section, all current treatments should be listed and when they were approved. Also, the references cited to support this section are not appropriated because most of them are not the original reports.
In line with the referee's criticisms, we recognized the title of this section as somewhat misleading. Indeed, our purpose was not to list the approved treatments for breast cancer therapy (it would also be a very complex topic that goes beyond the goal of this review), but to introduce in principle the different types of therapeutic approaches currently available for the treatment of this malignancy. In short, it is a preparatory section to the subsequent discussion. Therefore, to avoid misunderstandings, we have changed the section title to “Outline on the therapeutic approaches for BC treatment”. In addition, we have further highlighted the purpose of this section by inserting a preliminary sentence on the discussed topic. For this we thank the reviewer#3.
Section 3. This section is poorly organized and doesn’t make sense to be listed in this manuscript. It is very hard to follow why controlling apoptosis and cell death/cell survival can be prospective targeted therapies for breast cancer.
We carefully evaluated the reviewer’s considerations, but we believe this section, together with the following one (section 4), of central impact within the review context, since it focuses on the main deregulated cell death pathways and mechanisms playing a central role in several cancer pathologies, and promoting malignant cell survival. Indeed, cancer cells exhibit hallmarks of distinctive capabilities developed during tumorigenesis, including high proliferation rate and enhanced replicative potential, as well as apoptosis evasion and deregulated autophagy. Introducing and deepening these mechanisms - on which according to the most recent literature ruthenium-based drugs seem to act – we believe it is an important concern for the reading of the subsequent sections. Moreover, many anticancer drugs used both in clinic and/or in preclinical trials act by restoring typical tumour-dependent suppressed pathways, such as the apoptotic ones. This matches with the central topic of this special issue, where cancer metabolic rewiring promotes cell survival in response to tumorigenic molecular adaptations. However, in accordance with the reviewer's suggestions, we have attempted to review and simplify this section, as well as to further clarify the relationship between apoptosis regulation, cell survival and prospective targeted therapies.
Section 4. This section is very weak as well. The relationship between autophagy in breast cancer cells and “ruthenotherapy” is very weak.
As reported throughout the manuscript, to inhibit uncontrolled cell proliferation as in the case of metastatic tumors, the authors believe that one of the most promising research fields matches with the development of new chemotherapeutics capable of activating multiple mechanisms of cell death, where also autophagy can play a pivotal role. We agree with the reviewer that currently the relationship between “ruthenotherapy” and activation of autophagy may appear weak, but we feel this is due to the still little explored filed. However, some important correlations have been already reported concerning ruthenium-based treatments and activation of cytotoxic autophagy. By evaluation of the expression of some key autophagic markers, such as Beclin-1 and LC3-II, a significant upregulation in advanced preclinical models treated with KP1339 was clearly observed. In contrast, platinum-based derivatives did not show any pronounced effect, confirming findings of recent studies where cisplatin failed to induce autophagy whilst Ru-based drugs trigger cytotoxic autophagy (Wernitznig et al., Metallomics. 2019, 11(6), 1044-1048). Moreover, according to ruthenium complexes biological properties, these connections suggest the possibility for Ru-based candidate drugs of interaction with novel targets engaged in the regulation of autophagic processes, supporting our experimental data referred to in the review (Piccolo et al., Sci Rep. 2019, 9(1), 7006). However, based on the referee's comments, we have made an effort to improve also this section by making it more fluid and synthetic, and at the same time providing the basic information on which to graft the subsequent data concerning the effect of our Ru-containing nanosystems on the activation of autophagy.
Line 316, the advantageous properties of ruthenium complexes should be specified here.
Done
Line 415, specification of the multifactorial concerns (physical, chemical, and biological) would let the readers to easily follow the manuscript.
Ok, done.
Line 424, “both inorganic and organic” should be placed after “derivatives.”
Done
The data from Table 7 did not show significant selectivity of antiproliferative activity among different cell lines. Consequently, the section 8 doesn’t make sense to me. The authors should explain what kind of selectivity exists. The reasons for the selectivity should be illustrated.
We thank the reviewer for this note which allows us to further improve the manuscript and clarify some important concepts. As clearly indicated in the corresponding caption, Table 1 refers to the best results, in terms of IC50 values, obtained in human tumor models in vitro throughout preclinical studies. Table 1 does not include IC50 values ​​for each type of nanosystem in each type of cellular model used (for these data, several original papers by the authors have been cited). As also specified in the text, data showed in Table 1, which include outcome from cell models other than BC, have been used exclusively to demonstrate the antiproliferative efficacy of our nanosystems in cell lines of different histopathological origin and endowed with various replicative and metastatic potentials. Indeed, IC50 values ​​for cisplatin have been included under the same experimental conditions to facilitate a direct comparison of the antiproliferative capacities. Moreover, it can be noticed that most of the cellular models reported in Table 1 responsive to the ruthenium action in vitro are represented by different subtypes of mammary cells. In almost a decade of preclinical experimentation by screening on tumor cell panels, we have in fact proven that our Ru-containing nanosystems exhibit a constant and remarkable activity against all the BC lines tested, while their activity - although noteworthy in some cases (i.e., those reported in Table 1) - is more variable in cancer cells other than BC, showing less predictable positive results. Thus, as examples, HoUrRu complex lodged in both neutral and cationic nanosystems was active in WiDr human colorectal cancer cells, but it was not in additional cell lines included in the colon cancer screening panel such as HT29. Similarly, ToThyRu/DOTAP was active on glioblastoma LN229 cells, while the corresponding neutral nanosystem (ToThyRu/POPOC) was not, and both were inactive on SH-SY5Y human glioblastoma. Additionally, not least, the most remarkable antiproliferative activities we have detected throughout preclinical studies by using the more efficient DOTAP-based cationic nanosystems, have been observed in some subtypes of BC cells, such as MCF-7, but especially in their variants CG-5 and in T47D (IC50 less than 5 microM). In accordance, since the discovery of the anticancer potential of ruthenium-based complexes, several derivatives were reported as promising candidates for the treatment of BC, showing selectivity and efficacy in BC subtypes (Golbaghi et al., Molecules 2020, 25(2)). That said, to strengthen these concepts and matching to the reviewer's comments, we further emphasized these findings by illustrating in more detail the reasons behind the selectivity of action of AziRu-containing nanosystems. For this purpose, specific comments were added in both the sections “Preclinical validation of nucleolipid Ru-based nanoformulations in models in vitro” and “Selectivity and efficacy of nucleolipid Ru-based nanoformulations in BCC models”.
Round 2
Reviewer 2 Report
The authors have addressed all my concerns to the previous manuscript. I would like to recommend the publication of the manuscript in its current form.
Reviewer 3 Report
The authors have addressed all my concerns. I would like to recommend the publication of the manuscript in its current form.